# REUSING PRE-TRAINING DATA AT TEST TIME IS A COMPUTE MULTIPLIER

**Alex Fang**[2,3]    **Thomas Voice**[1]    **Ruoming Pang**[2]    **Ludwig Schmidt**[3]    **Tom Gunter**[2]
[1]Apple    [2] work done while at Apple    [3]Stanford

## ABSTRACT

Large language models learn from their vast pre-training corpora, gaining the ability to solve an ever increasing variety of tasks; yet although researchers work to improve these datasets, there is little effort to understand how efficient the pre-training apparatus is at extracting ideas and knowledge from the data. In this work, we use retrieval augmented generation along with test-time compute as a way to quantify how much dataset value was left behind by the process of pre-training, and how this changes across scale. We demonstrate that pre-training then retrieving from standard and largely open-sourced datasets results in significant accuracy gains in MMLU, Math-500, and SimpleQA, which persist through decontamination. For MMLU we observe that retrieval acts as a $\sim 5x$ compute multiplier versus pre-training alone. We show that these results can be further improved by leveraging additional compute at test time to parse the retrieved context, demonstrating a 10 percentage point improvement on MMLU for the public LLaMA 3.1 8B model. Overall, our results suggest that today's pre-training methods do not make full use of the information in existing pre-training datasets, leaving significant room for progress.

## 1    INTRODUCTION

Large language models (LLMs) have consistently improved performance by scaling pre-training compute (Hestness et al., 2017; Hoffmann et al., 2022; Kaplan et al., 2020). In parallel to scaling, researchers have also made significant efforts to improve both the architectures and the datasets used by these LLMs (Gu & Dao, 2023; Li et al., 2024; Raffel et al., 2020; Shazeer et al., 2017). While LLMs are able to solve an incredible number of tasks, in their current form they have several limitations. For example, they struggle with long-tail knowledge (Kandpal et al., 2023) and have limitations in their ability to generalize, such as for the reversal curse (Berglund et al., 2023). Additionally, they have been observed to have a log-linear scaling trend, meaning that larger amounts of compute are necessary to make the same gains at larger scales. To understand whether these limitations come from the quality of the datasets, it is important to explore whether further improvements can be unlocked by using them beyond pre-training.

Taking advantage of the effort put into creating these sophisticated pre-training datasets, we explore whether reusing them through retrieval at test time can further improve performance. Additionally, we test if supplementing with additional test-time compute results in further gains. Extra test-time compute can be applied naturally to retrieval augmented generation by running multiple trials for self-consistency while changing the retrieved documents across different trials. We apply this variety of techniques on a set of publicly available pre-training datasets, to measure the potential impact of the knowledge contained in them.

First, we pre-train models at various compute budgets and then use the same dataset for retrieval. We see that retrieval further benefits our models when evaluating on MMLU, Math-500, and SimpleQA, even though the models were pre-trained on the same data. We fit a power law to the base pre-training models across compute budgets and compare against the gain from retrieval, showing that while on average retrieval provides a $\sim 5x$ compute multiplier over pre-training when evaluating on MMLU, the effectiveness degrades with scale.

Next, we use additional test-time compute on the pre-training data with a combination of retrieval and self-consistency techniques. We evaluate our methods on a variety of downstream tasks that cover

multiple domains, requiring both knowledge and reasoning abilities. With a Llama 3.1 8B reader model, we use retrieval to achieve 74.0% on SimpleQA, as well as a 10.5 percentage point gain on MMLU, a 15.7 percentage point gain on MATH-500, and a 6.2 percentage point gain on GPQA.

Lastly, we analyze our findings to provide suggestions for further improving these datasets. We are able to relate performance gaps between controlled experiments back to specific stages of the dataset creation pipeline. Altogether, our work suggests that there is room for improving both pre-training datasets and learning methods using these datasets.

## 2 RELATED WORK

Classical pre-training scaling law studies established relationships for how loss falls with training compute, data, and parameters (Hestness et al., 2017; Kaplan et al., 2020; Hoffmann et al., 2022). More recent analyses factor in deployment costs in high-inference-demand settings Sardana et al. (2024).

Meanwhile, retrieval-augmented approaches externalize knowledge to non-parametric memory (Guu et al., 2020; Lewis et al., 2020; Ram et al., 2023), trading additional test time compute for improved performance on knowledge-intensive tasks. More recently, Shao et al. (2024) demonstrated that scaling up the datastore can reliably further improve performance on knowledge-intensive tasks, while Lyu et al. (2025) demonstrated that a compact subset of pre-training data can be used in a practical way with a minimal retrieval setup to improve performance on reasoning benchmarks.

Recent work (Brown et al., 2024; Snell et al., 2024) has shown that scaling test-time compute can be an efficient way of improving LLM performance. Specifically, they suggest parallelizing inference compute across trials, and sequentially iterating on a model's output. Results can then be aggregated with techniques like self-consistency (Wang et al., 2022; Chen et al., 2023) or verifiers. Retrieval is naturally suited to both as it can be parallelized across retrieved documents, while sequentially iterating over the search query to improve the retrieved documents.

Large-scale commercial "Deep Research" systems from companies like Google, OpenAI, and Perplexity likely apply all of the above by abstracting retrieval behind tool-use APIs and allowing the model direct control over the tools along with additional test-time compute.

In this paper, we first take inspiration from the classical pre-training scaling law studies, and begin to characterize the joint scaling of pre-training and simple retrieval given a fixed and identical data corpus. We also explore which forms of simple test-time compute allow us to most effectively leverage the retrieval stores, and whether these conclusions generalize to settings where the pre-training corpus differs from the retrieval datastore.

## 3 EXPERIMENTAL SETUP

### 3.1 DATASETS

We use the exact same datasets for both pre-training and retrieval. We include the standard webcrawl based large-scale datasets DCLM-baseline and FineWeb-edu (deduplicated versions for both), followed by more specialized sources. These include arXiv, peS2o, PubMed Central, Stack Exchange, and Wikipedia. Lastly, we include AlgebraicStack, AutoMathText, FineMath-3+, FineMath-4+, OpenWebMath, and StackMathQA to improve mathematics coverage. Additional information on the datasets such as token count and pre-training mixing ratios can be found in Table 1.

### 3.2 RETRIEVAL PIPELINE

We use Qwen3 Embedding 0.6B and Qwen3 Reranker 0.6B (Zhang et al., 2025a) as the embedding and reranker models. We use FAISS FlatIP (Johnson et al., 2019) for indexing to retrieve the top 100 documents per dataset (or shard of a dataset) before combining across all datasets through sorting by similarity score. This is equivalent to having all datasets in the same index and retrieving the top 100, but is more practical due to compute constraints. Then if applicable, we rerank the top 100 documents across all datasets to achieve our final document ordering. Lastly, we do retrieval augmented generation by concatenating and prepending the relevant documents before the question.

Table 1: Pre-training datasets used. Tokens refers to the actual size of the dataset, all of which is typically used during retrieval. Epochs refers to the number of epochs taken for the largest pre-training run. Epochs for smaller models are scaled down proportionally.

| Dataset | Tokens | Epochs |
|---|---|---|
| **Web Crawl Based** | | |
| DCLM-baseline (dedup) (Li et al., 2024) | 764.9B | 0.88 |
| FineWeb-edu (dedup) (Penedo et al., 2024) | 197.6B | 0.80 |
| **Additional Sources** | | |
| arXiv | 28.7B | 1.10 |
| peS2o (Soldaini & Lo, 2023) | 71.9B | 1.31 |
| PubMed Central | 22.5B | 2.72 |
| Stack Exchange | 11.8B | 7.98 |
| Wikipedia | 21.7B | 3.63 |
| **Math** | | |
| AlgebraicStack (Azerbayev et al., 2023) | 9.9B | 6.38 |
| AutoMathText (Zhang et al., 2025b) | 6.0B | 13.11 |
| FineMath-3+ (Allal et al., 2025) | 36.5B | 1.72 |
| FineMath-4+ (Allal et al., 2025) | 10.0B | 11.05 |
| OpenWebMath (Paster et al., 2023) | 13.5B | 2.91 |
| StackMathQA (Zhang, 2024) | 0.7B | 558.67 |

## 4 PRE-TRAINING EXPERIMENTS

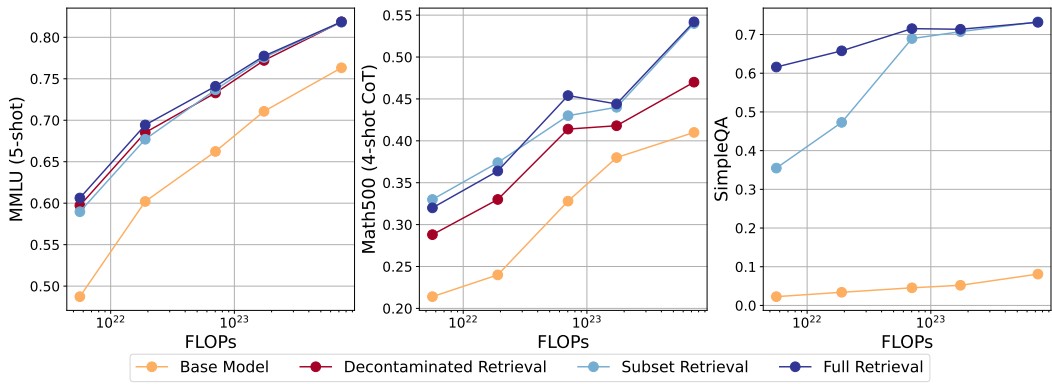

Figure 1: Retrieval on the pre-training dataset can substantially improve upon the performance of the base model. However, the exact benefit depends on the type of task.

We aim to measure the knowledge contained in pre-training datasets by first pre-training on it, then also retrieving (with reranker) on it during test time. In Figure 1 we measure performance on MMLU (Hendrycks et al., 2020), Math-500 (Lightman et al., 2023), and SimpleQA (Wei et al., 2024) across compute budgets, comparing the base model with retrieval on all our datasets, retrieval on a decontaminated version of all our datasets, and retrieval on a subset approximately equivalent to the unrepeated pre-training dataset. Overall, we find that retrieval can help on all three tasks, though to different degrees.

Retrieving from the full dataset leads to large gains on all tasks, but the model sees more data than the base model sees in pre-training. We account for this by retrieving from a subset similar to the unrepeated pre-training dataset, represented by the light blue lines in Figure 1. Interestingly, this achieves similar performance to retrieving on the full dataset for MMLU and Math-500. On the other hand, performance on SimpleQA is a piecewise function because SimpleQA depends heavily on

retrieving from Wikipedia, and the smaller two models only see a fraction of Wikipedia while the larger three models see at least one epoch of Wikipedia.

**Retrieval as a compute multiplier** Given the common use of MMLU as a proxy for pre-training quality, we use it to measure the effect of retrieval as a compute multiplier for the base model. We fit a bounded sigmoid function to the base model's MMLU performance as a function of FLOPs, and then measure the amount of pre-training compute needed to match each existing base model augmented with retrieval. In Table 2 we find that the average compute multiplier across the five models of different scales is 4.86x, and the geometric mean is 4.66x. However, this compute ratio decreases as the model scales, with retrieval providing only a 2.88x compute multiplier at the largest scale. While retrieval provides large compute savings, as the base model is scaled up, retrieval faces greater diminishing returns than just the base model. It is important to notice that retrieval does not provide a flat or strictly decreasing benefit across compute budgets. There is an initial increase in retrieval efficiency, suggesting that it benefits from better base models.

Table 2: We fit the base model performance to a sigmoid function with bounds to get $y = 0.25 + \dfrac{0.6907}{1 + \exp\left(-0.7968 \cdot (\log_{10}(x) - \log_{10}(2.48 \times 10^{22}))\right)}$, where 0.25 is the random baseline and 0.9407 is the maximum achievable accuracy (Gema et al., 2024). We use this equation to measure retrieval as a compute multiplier for the base model. The average compute ratio is 4.86x, the geometric mean is 4.66x, and the median is 4.74x.

| Compute Budget | Baseline MMLU | Retrieval MMLU | Compute for base to match retrieval | Compute Ratio |
|---|---|---|---|---|
| $5.64 \times 10^{21}$ | 0.4873 | 0.6063 | $2.98 \times 10^{22}$ | 5.28x |
| $1.90 \times 10^{22}$ | 0.6021 | 0.6943 | $1.36 \times 10^{23}$ | 7.17x |
| $7.04 \times 10^{22}$ | 0.6623 | 0.7410 | $3.34 \times 10^{23}$ | 4.74x |
| $1.74 \times 10^{23}$ | 0.7107 | 0.7775 | $7.35 \times 10^{23}$ | 4.23x |
| $7.34 \times 10^{23}$ | 0.7633 | 0.8186 | $2.11 \times 10^{24}$ | 2.88x |

**Decontamination** A common question is whether retrieval gains come from retrieving text containing exact overlap with the test data. We decontaminate the retrieved documents for MMLU and Math-500 through n-gram overlap with the questions, as detailed in Appendix B. In Figure 1, the red decontaminated retrieval line is close to the dark blue full retrieval line for MMLU, demonstrating that the gains are not attributable to simple contamination. Although Math-500 shows signs of more significant contamination, retrieving against a decontaminated training set still shows a very meaningful improvement over the baseline. We do note that our analysis shows that 14.1% of MMLU and 32.0% of Math-500 can be found in our commonly used open-source pre-training datasets, highlighting the importance of strictly decontaminated (or held out) evaluation sets for pre-training science. We choose to omit n-gram overlap decontamination analysis for SimpleQA due to the nature of the evaluation task.

## 4.1 LEARNING FROM PRE-TRAINING VS RETRIEVAL

In an effort to determine how retrieval can improve model performance, compared to scaling up model size and compute budget, we analyzed the accuracy of the trained models on MMLU, broken down into question categories. The results in Figure 2 show that across categories, the addition of retrieval gives a comparable boost in accuracy to that of a significant increase in pre-training compute budget.

Since retrieval involves a memory storage mechanism, we might expect it to provide most benefit for problems requiring good recall of facts, rather than reasoning abilities. However, in Table 3 we see that retrieval is a better compute multiplier for STEM than for humanities or social sciences, and in Figure 2 the gap between retrieval and base accuracy is also wider for STEM than for humanities. Such knowledge may be harder to absorb during pre-training, and in contrast to long tail facts in SimpleQA, it may be possible that retrieval expanding the context may also function as additional processing rather than just storage.

Table 3: Summary of pre-training vs retrieval compute ratios across MMLU categories. Values show how many times more compute the base model would need to match retrieval performance. Calculated using category-specific bounded sigmoids (min = 0.25; max: STEM 0.9544, Humanities 0.9377, Social 0.9575, Other 0.9114, All 0.9407).

| Compute Budget | STEM | Humanities | Social | Other | All |
|---|---|---|---|---|---|
| $5.64 \times 10^{21}$ | 6.82x | 2.69x | 3.42x | 9.80x | 5.28x |
| $1.90 \times 10^{22}$ | 10.23x | 3.33x | 4.42x | 13.78x | 7.17x |
| $7.04 \times 10^{22}$ | 5.23x | 2.57x | 4.07x | 8.77x | 4.74x |
| $1.74 \times 10^{23}$ | 5.01x | 2.44x | 3.43x | 7.53x | 4.23x |
| $7.34 \times 10^{23}$ | 3.52x | 1.55x | 2.22x | 6.48x | 2.88x |
| Average | 6.16x | 2.52x | 3.52x | 9.27x | 4.86x |
| Geometric Mean | 5.78x | 2.44x | 3.42x | 8.96x | 4.66x |
| Median | 5.23x | 2.57x | 3.43x | 8.77x | 4.74x |

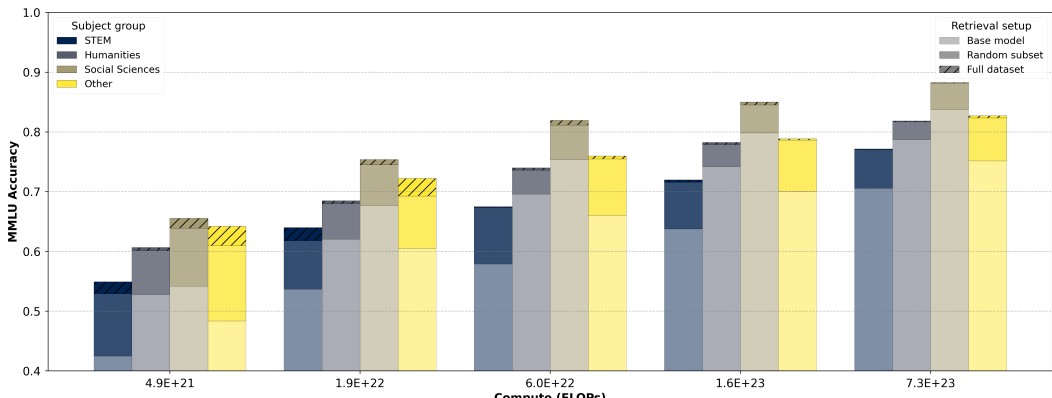

Figure 2: MMLU Breakdown by category of impact of retrieval addition and compute budget. Retrieval provides a strong lift, and the difference between retrieving from a random subset of the data store and the full set is small and diminishing with scale.

To investigate this surprising observation, we calculated the increase in MMLU accuracy provided by full retrieval, for different subject areas, with a 6.4B model. The top ten are shown in Table 4. We also include the corresponding accuracy increase provided by subset retrieval. Here we can see a mix of different subject types - including those that might be expected to require good knowledge recall, such as law and medicine, along with those that require computation, such as physics. A similar mixture of types of subjects may be seen for other model sizes, suggesting no strong correlation between subject type, model size, and the benefits of retrieval.

Table 4: Top ten categories ordered by change in MMLU accuracy after introduction of full retrieval

| Subjects | Impact of full retrieval | Impact of subset retrieval |
|---|---|---|
| Medical genetics | +21.1 | +19.5 |
| Miscellaneous | +19.2 | +17.8 |
| World religions | +18.5 | +19.6 |
| Philosophy | +17.9 | +17.1 |
| US foreign policy | +17.7 | +16.1 |
| International law | +17.0 | +17.3 |
| High school physics | +16.9 | + 5.7 |
| Virology | +16.9 | +15.2 |
| College physics | +16.8 | +14.2 |
| College medicine | +16.6 | +11.8 |

We also compared the MMLU answers between the 6.4B and 12.6B models, with and without retrieval, focusing on problems where the smaller base model gave an incorrect answer. We identified problems where the addition of retrieval corrected the answer, but the increase in model size did not, and vice versa. The main subjects of the former group were professional law, professional psychology, high-school macroeconomics, philosophy, and high-school mathematics. The main subjects of the latter group were professional law, professional psychology, moral scenarios, elementary mathematics, and high school statistics. Both groups have a mix of recall and reasoning problems, with much overlap. This suggests that there is not a strong bias for what kinds of problem retrieval can help with compared to increasing model size.

In total, increasing model size changed answers for 39.7% of MMLU problems, whereas adding retrieval changed answers for 28.1% of the problems. A similar analysis for MATH-500 showed that increasing model size changed answers for 39.7% of the problems, whereas adding retrieval changed answers for 28.7% of the problems. This suggests retrieval overall has less of an effect on model behavior than increasing model size. Furthermore, it suggests that in areas where retrieval does not help, the problem is more that the model ignores the additional context, rather than the additional context being misleading. Future work could involve encouraging models to better utilize retrieved context, possibly through prompt engineering or attention weighting.

## 5 TEST-TIME COMPUTE EXPERIMENTS

Table 5: Comparing baseline reader model performance against retrieval with a variety of test-time compute options. All evaluations use chain-of-thought reasoning. We use Llama 3.1 8B instruct as the reader model. MMLU results are reported as macro average over subjects. VR refers to using variance reduction techniques such as MMR and bagging.

| Method | MMLU STEM | MMLU Humanities | MMLU Social | MMLU Other | MMLU All |
|---|---|---|---|---|---|
| Baseline | 67.3 | 71.5 | 76.6 | 73.0 | 71.6 |
| w/ self-consistency | 72.3 | 74.8 | 79.3 | 76.4 | 75.3 |
| w/ retrieval | 73.6 | 74.6 | 81.8 | 77.6 | 76.6 |
| w/ reranker | 73.7 | 76.3 | 83.4 | 79.2 | 77.7 |
| w/ reranker + self-cons. | 78.7 | 78.7 | 85.8 | 81.9 | 81.0 |
| w/ reranker + self-cons. + VR | 80.2 | 79.5 | 87.4 | 82.3 | 82.1 |

Table 6: Continuation of Table 5. GPQA, and Math-500 results are over 10 trials.

| Method | SimpleQA | Math-500 | GPQA Bio. | GPQA Chem. | GPQA Phys. | GPQA All |
|---|---|---|---|---|---|---|
| Baseline | 1.5 | 48.7 | 46.2 | 26.4 | 28.3 | 30.6 |
| w/ self-consistency | N/A | 55.9 | 46.3 | 28.1 | 28.4 | 31.4 |
| w/ retrieval | 65.7 | 56.7 | 45.1 | 27.3 | 34.0 | 33.2 |
| w/ reranker | 74.0 | 56.8 | 46.7 | 28.6 | 36.0 | 34.8 |
| w/ reranker + self-cons. | N/A | 64.3 | 48.5 | 30.1 | 36.8 | 36.1 |
| w/ reranker + self-cons. + VR | N/A | 64.4 | 49.7 | 29.6 | 38.3 | 36.8 |

Knowing the limitations of learning with just pre-training, we attempt to better quantify the knowledge contained in these datasets by applying additional test-time compute on top of retrieval. If the model is able to answer a question with retrieval and test-time compute, the knowledge required to do so is likely in the dataset. In this section, we use Llama 3.1 8B instruct (Grattafiori et al., 2024) as the reader model due to its performance relative to its size, making it more practical to apply test-time compute. We then augment it with retrieval as described in Section 3, and parallel inference with majority voting to select an answer (self-consistency). In addition to evaluating on MMLU, SimpleQA, and Math-500, we also evaluate on GPQA (Rein et al., 2024).

Table 5 and Table 6 show that the effects of self-consistency and retrieval are additive across all tasks, with the exception of SimpleQA where self-consistency does not help because it is purely a factuality benchmark. Perhaps surprisingly, the two techniques help generally across all other tasks and sub-tasks, with little hint of specialization. Additionally, reranking seems to give a consistent boost on top of retrieval across tasks. Lastly, we take advantage of retrieving multiple documents and parallelizing trials by using older techniques like MMR (Carbonell & Goldstein, 1998) to increase diversity, and bagging (Breiman, 1996) (randomizing over a subset of documents) to reduce variance; these techniques give a further performance boost for MMLU and GPQA.

If we view retrieval as a tool for the LLM, then our methods use test-time compute to improve the tool itself. This contrasts with self-consistency by itself, which parallelizes the model without additional enhancement, as well as with deep research, which in addition to parallelizing also uses test-time compute to use the tool for longer rather than to upgrade it. Retrieval is our vehicle through which we can put in additional compute, and do it in a data driven way.

## 5.1 LEARNING FROM TEST-TIME COMPUTE VS PRE-TRAINING

Table 7: Compute efficiency gains for each method relative to baseline performance. Values represent how many times more compute the baseline model would need to achieve the same performance as each method. Calculated using fitted sigmoid equations for each MMLU category.

| Method | MMLU STEM | MMLU Humanities | MMLU Social | MMLU Other | MMLU All |
|---|---|---|---|---|---|
| Baseline | 1.00× | 1.00× | 1.00× | 1.00× | 1.00× |
| w/ self-consistency | 2.62× | 1.93× | 1.63× | 2.21× | 2.10× |
| w/ retrieval | 3.42× | 1.85× | 2.70× | 3.02× | 2.78× |
| w/ reranker | 3.49× | 2.67× | 3.87× | 4.72× | 3.56× |
| w/ reranker + self-cons. | 10.74× | 4.65× | 7.18× | 11.34× | 8.14× |
| w/ reranker + self-cons. + VR | 15.72× | 5.68× | 11.66× | 13.15× | 11.10× |

We can take the MMLU sigmoid fit from Section 4 to analyze Table 5. Although Llama 3.1 8B is trained at a much higher tokens per parameter ratio than the models in Section 4, the sigmoid fit could still be reasonable because calculating the compute ratio between retrieval (with reranker) and base performance is within reason for MMLU (All). Additionally, estimates using our previous sigmoid fits would be a lower bound for compute multipliers because of diminishing returns at higher tokens per parameter counts.

Despite the different pre-training dataset and Llama being significantly overtrained, we see that retrieval (with reranker) still functions as a 3.56x compute multiplier, similar to what would be expected of a model with the same MMLU base accuracy in our pre-training setup in Section 4. Though we do not know the details of the Llama 3.1 pre-training dataset, it is likely that there is substantial overlap with the data we are retrieving from. Table 7 shows that, altogether, our methods provide at least an 11x compute multiplier over the pre-trained baseline.

We also see that the different test-time methods learn or utilize data differently from pre-training, as the multipliers are different across the categories. Even the different test-time methods have different behaviors, as both self-consistency and retrieval favor STEM and other, while the lift of reranker over retrieval favors humanities, social sciences, and other.

## 5.2 A CONNECTION BETWEEN RETRIEVAL AND CONSISTENCY

While the previous results in this section demonstrate that self-consistency is a powerful tool for improving performance, it can also be used as an analytical tool for retrieval. As displayed in Appendix H, we can apply self-consistency on each individual document and rerank the documents with it. In Table 8 we see that inter-document consistency selects better top-1 documents than the reranker. However, this technique also requires a multiplicative number of additional trials, as previously we ran a fixed number of trials on all documents combined, but now we are doing it per document. We leave to future work ways to distill self-consistency into a more efficient reranker.

Table 8: Inter-document consistency can act as a stronger reranker (k=1) than standard rerankers from Zhang et al. (2025a); however, it requires calling the reader model many more times and is not compute efficient when compared to self-consistency on all documents at once.

| Reranker | MMLU STEM | MMLU Humanities | MMLU Social | MMLU Other | MMLU All |
|---|---|---|---|---|---|
| Qwen3 Reranker 0.6B | 71.2 | 72.9 | 77.7 | 74.1 | 73.7 |
| Inter-doc consistency | 75.4 | 75.4 | 82.7 | 78.1 | 77.6 |

## 6 ADDITIONAL ANALYSIS

### 6.1 BETTER PRE-TRAINING DATASETS ARE NOT NECESSARILY BETTER RETRIEVAL DATASETS

Table 9: While FineWeb-edu is worse than DCLM when measuring pre-training performance, it is just as good if not slightly better for retrieval. Retrieval is on top of Llama 3.1 8B instruct with k=10. Pre-training numbers are 8B models trained for 1T tokens, as reported by Su et al. (2024).

| Dataset | Pre-training MMLU | Retrieval MMLU | Retrieval w/ reranker MMLU |
|---|---|---|---|
| DCLM | 53.4 | 74.5 | 76.4 |
| FineWeb-edu | 42.9 | 75.2 | 76.6 |

Our retrieval datasets were built for the purpose of pre-training, which raises the question of whether better pre-training datasets make for better retrieval datasets. Table 9 suggests that this is not necessarily the case, as FineWeb-edu is worse than DCLM for pre-training, but is as good if not slightly better for retrieval. While DCLM contains more tokens than FineWeb-edu, prior work (Muennighoff et al., 2023b; Fang et al., 2025) would suggest that this is not the reason for the gap in pre-training performance, while the size advantage should not be harmful for retrieval. We leave to future work how to determine the qualities that make a dataset good for pre-training or retrieval specifically.

### 6.2 IMPORTANCE OF EXTRACTION AND CRAWLING

Table 10: Text extraction and crawling are important for creating good datasets. We vary the extraction done on top of Wikipedia, as well as how we expand the datastore. Retrieval uses reranker and is on top of Llama 3.1 8B instruct with k=6.

| Dataset | SimpleQA |
|---|---|
| Wikimedia Nov. 2023 | 55.4 |
| OLM June 2025 | 59.1 |
| Custom June 2025 | 69.0 |
| Custom + All Sources | 73.7 |
| Custom + Golden Links | 85.2 |

We investigate the importance of earlier stages of dataset creation through a case study of retrieval for SimpleQA. As constructed, over 70% of the answers in SimpleQA can be found on Wikipedia. However, many works use out-of-date or pre-extracted versions of Wikipedia. This can lead to missing data, especially when the crucial piece of information comes from specialized elements.

We compare Wikimedia (Nov. 2023) (Wikimedia-Foundation) and OLM (June 2025) (Thrush et al., 2022), two of the most popular Wikipedia extractions on HuggingFace, against a custom extracted Wikipedia (June 2025) described in Appendix C. Qualitatively, we find that existing extractions often fail to extract elements like bullet points, tables, and info boxes. Table 10 quantitatively demonstrates this with up to a 13.6 percentage point difference in SimpleQA performance by simply changing the version of the Wikipedia dataset.

Next, we compare expanding the retrieval datastore by adding our other sources against adding non-Wikipedia golden links provided by SimpleQA. Table 10 shows that there is an 11.5 percentage point difference, and we find that only a small fraction of the non-Wikipedia golden links are present in CommonCrawl. This suggests that open-source datasets could be further improved at the web crawling stage.

### 6.3 Robustness when scaling retrieval data

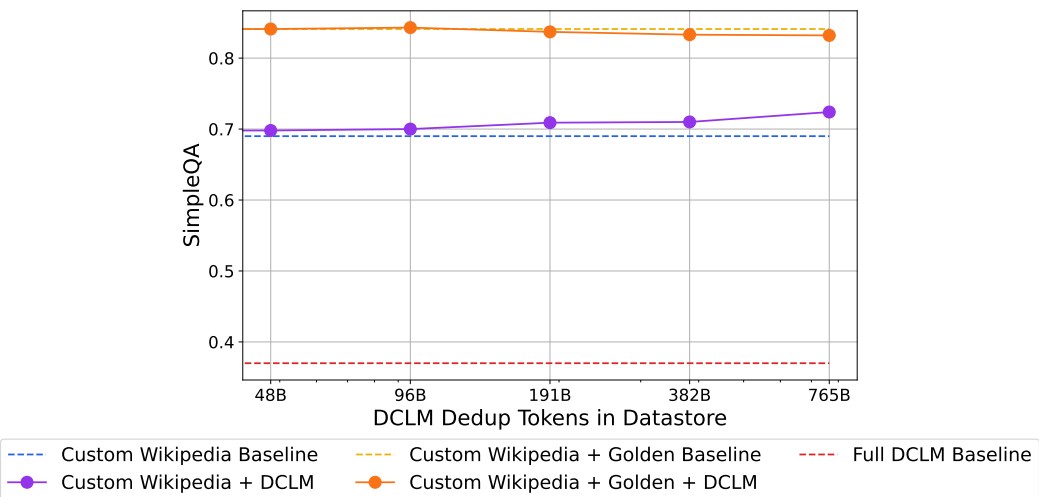

Figure 3: For SimpleQA, our retrieval system is fairly robust to scaling the retrieval datastore, even if the new data does not contain useful information. Our custom Wikipedia contains 22B tokens, and additional DCLM data helps a little, or when also starting with additional golden link data, hurts only a little.

Given that most of SimpleQA can be solved with Wikipedia and the additional provided golden links, we investigate the effect of additional web crawl data on the retrieval system. In Figure 3 we see that the additional data has only a small distracting effect, as the accuracy on SimpleQA stays close to the baseline. However, we acknowledge that SimpleQA measures factual knowledge, and the scaling effect may be different for reasoning tasks.

## 7 Future work

We have shown that pre-training does not fully utilize all the knowledge contained within today's open-source pre-training datasets. This would suggest that there are still many algorithmic improvements left to explore. Additionally, in our process of analyzing data quality, we have also shown that there is room for improving datasets, at the very least in terms of crawling and extraction.

Within this work, we explore a limited number of simple test-time techniques on a limited set of evaluations. However, it is quite likely that applying advanced techniques like query rewriting, test-time training, and reinforcement learning for retrieval will further boost the performance on the same datasets (Hardt & Sun, 2023; Ma et al., 2023). We also believe that these findings apply to even broader domains. Initial results in Appendix G suggest that retrieving from pre-training datasets also benefits code generation.

Beyond improvements, we would also like to better understand how data is used during pre-training. Our measurement of retrieval as a compute multiplier parallels that of well-tuned Mixture-of-Experts models (Clark et al., 2022). Additional exploration in this area could uncover whether these two methods have significant overlap in terms of data usage.

ACKNOWLEDGMENTS

We would like to thank Sewon Min, Rulin Shao, and Anikait Singh for helpful conversations and feedback at various stages of the project.

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

## A   PRE-TRAINING DETAILS

We follow a recipe very similar to that open-sourced as the "Honeycrisp" model series in the Axlearn training framework Lee et al. (2025).

We train each dense decoder model for around 20 tokens per parameter, and follow a cosine-with-linear-warmup learning rate schedule with a peak learning rate of 1e-2, decaying to 0.01 of the peak lr. As described in the "Honeycrisp" model definitions, we use a muP-style parameterization to achieve learning rate transfer as we scale up the models. The model architecture is similar to the LLaMA series, with Swi-GLU FFNs, RoPE positional encodings, and Grouped Query Attention (GQA) using a key/value-to-query ratio of 1:8.

| Compute Budget | Paramters | Tokens |
|---|---|---|
| $5.64 \times 10^{21}$ | 6.4B | 147B |
| $1.90 \times 10^{22}$ | 12.6B | 252B |
| $7.04 \times 10^{22}$ | 23.3B | 503B |
| $1.74 \times 10^{23}$ | 36.8B | 786B |
| $7.34 \times 10^{23}$ | 77.8B | 1573B |

## B   DECONTAMINATION

We perform n-gram decontamination against our test sets using n-grams in token-space (according to our tokenizer). We drop entire documents on collision with a single 16-gram from the MMLU test-set or 26-gram for Math-500. Through visual examination we found that shorter n-gram overlaps were too aggressive (noting that our tokenizer e.g. splits numbers into single digits).

## C   CUSTOM EXTRACTION

We implement a custom HTML extraction pipeline and apply it to all pages from the Wikipedia domain found in our general web-crawl. Specifically, we first apply a lightweight pre-processing step to remove script, style, unmatched meta tags, HTML comments, links, and images. We then use the ReaderLM-v2 (Wang et al., 2025) to extract the coarsely simplified HTML into structured plain-text.

We note that this approach improves on recall (especially for tables and some information-boxes) over publicly available Wikipedia extractions, including the one recently provided by the Wikimedia organization Wikimedia.

## D   DETOKENIZE SUBSET VS RANDOM SUBSET

Table 11: In Figure 1 we show the effect of retrieving from a subset similar to that seen in pre-training for that compute budget. Specifically, we take a random subset that is the same size of what is seen during pre-training for each source. In this table, we compare at the smallest $5.64 \times 10^{21}$ compute budget between random subset and detokenizing the exact pre-training data and using that for retrieval. The results are fairly similar, and the larger gap in Math-500 may be attributed to randomness or contamination. Note that this gap will shrink as the compute budget increases because the two datasets will have increasing overlap.

| Data | MMLU | Math-500 | SimpleQA |
|---|---|---|---|
| Random subset | 59.0 | 32.0 | 35.5 |
| Exact subset | 58.6 | 26.8 | 33.9 |

# E FITS BY MMLU CATEGORIES

Table 12: Pre-training vs. retrieval compute ratios for STEM. Sigmoid fit (min=0.25, max=0.9544): $y = 0.25 + \dfrac{0.7044}{1 + \exp\big(-0.7351 \cdot (\log_{10}(x) - 22.9965)\big)}$. Average compute ratio is 6.16×, geometric mean is 5.78×.

| Compute Budget | Baseline MMLU | Retrieval MMLU | Compute for base to match retrieval | Compute Ratio |
|---|---|---|---|---|
| $5.64 \times 10^{21}$ | 0.4247 | 0.5493 | $3.85 \times 10^{22}$ | 6.82× |
| $1.90 \times 10^{22}$ | 0.5367 | 0.6399 | $1.94 \times 10^{23}$ | 10.23× |
| $7.04 \times 10^{22}$ | 0.5788 | 0.6749 | $3.68 \times 10^{23}$ | 5.23× |
| $1.74 \times 10^{23}$ | 0.6376 | 0.7197 | $8.71 \times 10^{23}$ | 5.01× |
| $7.34 \times 10^{23}$ | 0.7056 | 0.7705 | $2.58 \times 10^{24}$ | 3.52× |

Table 13: Pre-training vs. retrieval compute ratios for Humanities. Sigmoid fit (min=0.25, max=0.9377): $y = 0.25 + \dfrac{0.6877}{1 + \exp\big(-0.8008 \cdot (\log_{10}(x) - 22.1259)\big)}$. Average compute ratio is 2.52×, geometric mean is 2.44×.

| Compute Budget | Baseline MMLU | Retrieval MMLU | Compute for base to match retrieval | Compute Ratio |
|---|---|---|---|---|
| $5.64 \times 10^{21}$ | 0.5277 | 0.6013 | $1.51 \times 10^{22}$ | 2.69× |
| $1.90 \times 10^{22}$ | 0.6205 | 0.6847 | $6.33 \times 10^{22}$ | 3.33× |
| $7.04 \times 10^{22}$ | 0.6961 | 0.7398 | $1.81 \times 10^{23}$ | 2.57× |
| $1.74 \times 10^{23}$ | 0.7420 | 0.7788 | $4.24 \times 10^{23}$ | 2.44× |
| $7.34 \times 10^{23}$ | 0.7871 | 0.8169 | $1.14 \times 10^{24}$ | 1.55× |

Table 14: Pre-training vs. retrieval compute ratios for Social Sciences. Sigmoid fit (min=0.25, max=0.9575): $y = 0.25 + \dfrac{0.7075}{1 + \exp\big(-0.9563 \cdot (\log_{10}(x) - 21.9772)\big)}$. Average compute ratio is 3.51×, geometric mean is 3.42×.

| Compute Budget | Baseline MMLU | Retrieval MMLU | Compute for base to match retrieval | Compute Ratio |
|---|---|---|---|---|
| $5.64 \times 10^{21}$ | 0.5419 | 0.6555 | $1.93 \times 10^{22}$ | 3.42× |
| $1.90 \times 10^{22}$ | 0.6770 | 0.7538 | $8.39 \times 10^{22}$ | 4.42× |
| $7.04 \times 10^{22}$ | 0.7538 | 0.8192 | $2.86 \times 10^{23}$ | 4.07× |
| $1.74 \times 10^{23}$ | 0.7983 | 0.8501 | $5.97 \times 10^{23}$ | 3.43× |
| $7.34 \times 10^{23}$ | 0.8375 | 0.8828 | $1.63 \times 10^{24}$ | 2.22× |

Table 15: Pre-training vs. retrieval compute ratios for Other. Sigmoid fit (min=0.25, max=0.9114): $y = 0.25 + \dfrac{0.6614}{1 + \exp\big(-0.8008 \cdot (\log_{10}(x) - 22.2759)\big)}$. Average compute ratio is 9.27×, geometric mean is 8.96×.

| Compute Budget | Baseline MMLU | Retrieval MMLU | Compute for base to match retrieval | Compute Ratio |
|---|---|---|---|---|
| $5.64 \times 10^{21}$ | 0.4834 | 0.6418 | $5.53 \times 10^{22}$ | 9.80× |
| $1.90 \times 10^{22}$ | 0.6046 | 0.7222 | $2.62 \times 10^{23}$ | 13.78× |
| $7.04 \times 10^{22}$ | 0.6601 | 0.7598 | $6.17 \times 10^{23}$ | 8.77× |
| $1.74 \times 10^{23}$ | 0.7006 | 0.7882 | $1.31 \times 10^{24}$ | 7.53× |
| $7.34 \times 10^{23}$ | 0.7516 | 0.8271 | $4.76 \times 10^{24}$ | 6.48× |

## F MATH-500 CHECKER FOR USC

Table 16: Math-500 answers are open-ended so we use universal self-consistency (USC) (Chen et al., 2023) in Table 6 with the reader model itself as the checker (Llama 3.1 8B). Here, we compare this against using GPT-4.1 mini as the USC checker model.

| Method | Llama 3.1 8B checker | GPT-4.1 mini checker |
|---|---|---|
| Baseline | 48.7 | N/A |
| w/ self-consistency | 55.9 | 62.2 |
| w/ retrieval | 56.7 | N/A |
| w/ reranker | 56.8 | N/A |
| w/ reranker + self-cons. | 64.3 | 69.7 |
| w/ reranker + self-cons. + VR | 64.4 | 71.8 |

## G LIVECODEBENCH RESULTS

Table 17: Retrieving from the python portions of the Stack v2 (Lozhkov et al., 2024) and CommitPack (Muennighoff et al., 2023a) to augment generation for LiveCodeBench Code Generation (Jain et al., 2024).

| Model | Baseline | Retrieval (k=3) |
|---|---|---|
| gpt-4o-2024-08-06 | 0.3793 | 0.4276 |

# H    INTER-DOCUMENT CONSISTENCY

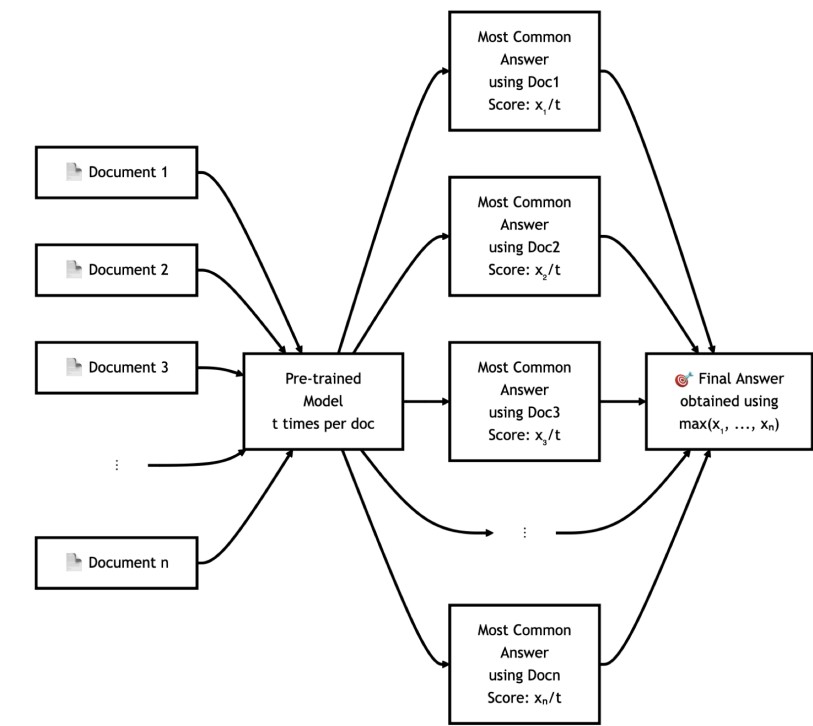

Figure 4: Inter-document consistency can be used to analyze retrieval and consistency. We apply self-consistency on generating while retrieving from individual documents, and select the answer from the most self-consistent document.

