# OpenReview forum: "Reusing Pre-Training Data at Test Time is a Compute Multiplier"
_ICLR.cc/2026/Conference — ICLR 2026 Poster_

### Official Review · Reviewer_Kf1X · 2025-11-01

**Soundness:** 3
**Presentation:** 3
**Contribution:** 3
**Rating:** 6
**Confidence:** 3

**Summary:**

This paper explores an interesting idea: reusing pre-training data at test time through retrieval augmentation to boost the performance of large language models. The authors demonstrate that this method acts as a "compute multiplier," meaning it achieves performance gains that would otherwise require significantly more computational resources for pre-training alone. They test this on several benchmarks like MMLU, Math-500, and SimpleQA, showing that retrieval, especially when combined with techniques like self-consistency and reranking, leads to substantial improvements. The key takeaway is that current pre-training methods don't fully squeeze all the juice out of the vast datasets they use, and test-time retrieval is a powerful way to unlock that leftover potential.

**Strengths:**

1.  The core concept of treating test-time retrieval as a "compute multiplier" is a really neat and effective way to frame the value of retrieval-augmented generation.

2.  The experiments are quite thorough, covering multiple datasets and model scales, which provides a solid empirical foundation for the claims.

3.  The analysis of how retrieval impacts different MMLU categories (like STEM vs. humanities) offers some cool insights into where this technique helps the most.

**Weaknesses:**

1.  The study focuses on models pre-trained and retrieved from the *same* dataset, which might not always be the case in real-world applications.

2.  While the gains are impressive, the paper notes that the effectiveness of retrieval as a compute multiplier diminishes as the base model size increases, which could limit its long-term utility for future, even larger models.

3.  The techniques explored, like retrieval and self-consistency, add significant computational overhead at inference time, and the trade-offs aren't discussed in much detail.

**Questions:**

1.  You showed that retrieval from the pre-training corpus is very effective. How do you think this approach would perform if the retrieval database was a completely different, specialized corpus that the model has never seen before? Would the "compute multiplier" effect still hold?

2.  In your analysis (Table 4), the performance lift from subset retrieval is surprisingly close to that of full retrieval for many MMLU subjects. Why do you think a smaller, targeted subset of the data provides such a comparable boost to having the entire dataset available for retrieval?

---

> ### Author Response · Authors · 2025-12-03
>
> Thank you for your time reviewing our work.
>
> The study intentionally focuses on models pre-training and retrieving on the same dataset because we want to measure how well pre-training learns from the dataset. While real-world applications may retrieve from smaller specialized datasets, our work would also suggest that the application may still benefit from retrieving from large scale datasets, e.g. retrieving from large scale code data may benefit specific coding tasks.
>
> As acknowledged by the reviewer and our work, the results diminish with model size increase, but even for our largest dense model the gains are non-trivial. Additionally, many large models are now MoEs rather than dense, and it would be of interest to see if MoEs have different retrieval trends.
>
> We do not discuss in detail the additional computational overhead because the focus is on measuring the utility of the data in relation to pre-training. However, the reviewer raises a great point that this matters a lot if this system were to be deployed in practice.
>
> If we substitute retrieving from pre-training data with a specialized corpus that the model has never seen before, the effect would depend on how good the model already is on the evaluation task, the nature of the task, as well as how the compute multiplier is calculated. Focusing more on the last point, if the compute multiplier is comparing retrieval on the specialized dataset against scaling standard pre-training data, the compute multiplier will probably be even higher; but a different example would be comparing it against fine-tuning on new data similar to the specialized corpus, which would be less effective than the effect seen in our study. One example is the very large effect of retrieving from Wikipedia on SimpleQA (despite seeing it in pre-training) because it is like a specialized corpus for that evaluation.
>
> The subset retrieval is actually on a random subset equivalent in size to the data seen during pre-training, rather than a targeted subset. We believe the effect is comparable because these datasets are already very large and there are diminishing returns on scaling.

---

### Official Review · Reviewer_xk27 · 2025-11-02

**Soundness:** 2
**Presentation:** 3
**Contribution:** 3
**Rating:** 4
**Confidence:** 3

**Summary:**

The authors pose the question of how well pretrained LLMs can extract information from their pretraining data. To study this, they propose using a RAG setup where the pretraining corpus itself is used for retrieval at test time. Through a broad set of experiments, they report notable accuracy gains on MMLU, MATH-500, and SimpleQA, and argue that these improvements indicate that current pretraining does not fully exploit the information in the data. They further claim that retrieval acts as a compute multiplier, a ~5x gain on MMLU, since, under a fixed FLOP budget, their RAG approach achieves the same accuracy as the base model with fewer FLOPs. They also show that adding test-time scaling via self-consistency leads to additional improvements.

**Strengths:**

- The paper is well written, and the methodology is clearly explained.
- The central question of "how much of the pretraining data is left unexploited after pretraining" is interesting and well motivated, and can help understanding the strengths and limitations of current LLMs.
- The authors conduct a substantial number of experiments, systematically testing different aspects of their claims.
- The pretraining experiments of Section 4, particularly those after decontamination, indeed suggest that pretraining alone does not always capture all patterns present in the pretraining corpus.

**Weaknesses:**

- I am not sure if the compute multiplication arguments are fair. To my understanding, the authors do not take into account the FLOPs associated with the retrieval and reranking components of their RAG pipeline when comparing against the base model’s pretraining FLOPs. As a result, the “under fixed compute budget” claim becomes somewhat misleading, and the reported multipliers (sometimes ~10x) are possibly inflated.
- In addition to my previous argument, I am not fully convinced that the main message of the paper is justified. While the results do show consistent improvements with retrieval, it is not clear whether these gains truly reflect “unused” information from pretraining, or simply the effect of providing additional access to relevant text at test time. Thus, I believe that more experiments in a more controlled setting would be necessary to support this hypothesis.

**Questions:**

- Do you believe that the results would change drastically if the RAG corpus was different than the pretraining corpus?

---

> ### Author Response · Authors · 2025-12-03
>
> Thank you for your time reviewing our work.
>
> While we do not include inference time compute as part of our compute multiplier calculation, we do this because we aim to measure the knowledge contained within the dataset that is lost by pre-training being an imperfect learning algorithm. Nonetheless, both the embedding and reranker models are 0.6B parameter models, so the effect would be insignificant, especially at the larger scales—a forwards pass over the training corpus with 0.6B is ~1.2e9 FLOPs per token in the corpus, vs at our largest model size (77B) 462e9 training flops per token—so < 1%. A bigger concern could be the multiple inference passes done for consistency, but it is difficult comparing pre-training and inference compute because the tradeoff is a function of the volume of inference queries over the lifetime of the model.
>
> Additionally, we agree with the review that the gains could be due to providing additional access to relevant text at test time. However, this would imply that this knowledge exists within the dataset, which was not fully acquired during pre-training. As the space for such exploration ranges from architectural improvements to distillation, we leave this for future work.
>
> Lastly, how a different RAG corpus would affect results would depend on the evaluation metric as well as the difference between the pre-training data and the new RAG data. If the new RAG dataset is similarly large scale and general, we believe the results would be pretty similar, as suggested by the comparison between using just DCLM and just FineWeb-edu. Another example is that it is likely Llama 3.1 used a different pre-training dataset but perhaps of similar quality to the one we used, yet still benefits from retrieval and additional test-time compute. On the other hand, a small specialized RAG dataset would help a lot more than mixing it once in pre-training, as demonstrated by Wikipedia and SimpleQA.

---

### Official Review · Reviewer_EuQe · 2025-11-02

**Soundness:** 3
**Presentation:** 2
**Contribution:** 2
**Rating:** 6
**Confidence:** 3

**Summary:**

This submissions studies the benefit of retrieval augmented generation in terms of how it compares to increased pre-training. It shows that retrieval potentially yields benefits comparable to significant multiples of (pre-)training time. A number of experiments analyze various related aspects of pre- and test-time training of LLMs.

**Strengths:**

The paper argues an interesting and highly-valuable point: well-done retrieval can comparatively benefit performance much more than additional generic model training. The idea of presenting gains in terms of "compute multipliers" is convincing (although the underlying simple sigmoid model deserves more attention) . The paper also provides evidence of additive benefits with other test-time procedures, which is promising. There is a massive amount of experiments here, supporting these findings and further analyses. The choice of datasets is clear and reasonable.

**Weaknesses:**

The main weakness of the paper is the general clarity re. exactly what is done. One blatant example is the Experimental Setup section, esp. 3.2: The short paragraph provides a straightforward and high-level description of the retrieval process, but no description of how the retrieved documents are used -- presumably a RAG-style process, but absolutely no detail is provided. Given that this is mainly an experimental paper, it is absolutely necessary that experimental details are provided, at minima with references, so that the reader knows what is done. There is also some confusion about what model(s) was/were used. The text mentions at various places 6.4B models, 8B models and 12.6B models -- it is not always clear which was used where (and why).

There is a dizzying amount of experiments done in Section 4, 5 and 6, providing a lot of experimental evidence. Unfortunately, the amount of description for each experiment is minimal, often only a couple of sentences with tables or graphs that are poorly described. It is often unclear what is reported (e.g. what metric is used). And no error bars or assessment of experimental uncertainty are shown.

Several interesting findings arise from the many paper's experiments. However some sound more like speculative guesses, e.g "might require logical/abstract reasoning" (l.254-255), "the model ignores additional context, rather than additional context being misleading"(??, l.284-285), "it is likely that there is substantial overlap" (l.362-363). Even the overall message from the paper, that pre-training does not make "full use" of the data, is unconvincing and not unambiguously supported by the data. For example, observations may just be the unsurprising consequence of leveraging test-specific data versus using a general-purpose model out-of-the-box.

**Questions:**

Gains for SimpleQA are striking (Fig. 1) -- so large that they don't really fit the "compute multiplier" narrative. What is going on with retrieval on that benchmark?

How were the five compute budget points chosen? Also it was not 100% clear whether model size and architecture are kept fixed as compute grows.

When retrieval is done, is it done on the full dataset, or only the part used for pre-training?

What exactly is "subset retrieval"? The description on l.159-160 is not sufficient.

l.161: "a piecewise function" -- this by itself is not very informative. Arguably most curves on Math500 are also "piecewise".

How are the variance reduction techniques used in the context of this paper (l.329-330)?

This is a matter of taste and author choice of course, but there is no conclusion beyond future work?

---

> ### Author Response · Authors · 2025-12-03
>
> Thank you for your time reviewing our work.
>
> We would like to clarify the details as the reviewer has requested. We are doing retrieval augmented generation, where the retrieved documents are concatenated then appended before the question. The models used in Section 4 span multiple sizes (sharing architecture, but scaling depth/width to maintain Chinchilla compute optimality at each FLOP budget), and are described in Appendix A, and inference is done using HuggingFace generation. In Section 4.1 we mention either the model size or compute budget (which are interchangeable) for each set of analyses, which typically is the 6.4B model unless comparing across sizes, which then also includes the 12.6B model. Section 5 only uses the Llama 3.1 8B model, which is served using vLLM. We are happy to provide any additional clarifications, and will endeavour to improve clarity in the body of the paper.
>
> We would also like to acknowledge that the lines mentioned by the reviewer are a bit speculative, and we will change the first two, but speculation about Llama pre-training data may be reasonable given that the bulk of pre-training data does come from web crawl, supplemented by common sources like Wikipedia and StackExchange. Although the filtering may be different, there is still likely substantial overlap. Additionally, we believe that our claim is substantiated by the data because we are using the same data at both pre-train and test-time and showing there is a gap in performance, meaning that there is knowledge in the data not learned by pre-training. The “test-specific” data was seen during pre-training during the Section 4 experiments.
>
> To address the additional questions:
>
> SimpleQA is a fact-based QA evaluation designed to be adversarial to frontier models at the time. The retrieval gains are especially large because a lot of the answers can be found in these datasets, but the questions were picked so that models have a hard time learning them at training time.
>
> The compute budgets were chosen to span a variety of model sizes while keeping tokens per parameter fixed at around 20 (i.e. Chinchilla optimal). Therefore model width and depth grow with compute budget, although the architecture remains consistent. Additional details can be found in Appendix A.
>
> Retrieval is done on the full dataset unless specified, which happens in Figure 1 for subset retrieval and decontaminated retrieval. Subset retrieval takes a random subset that is the same size of what is seen during pre-training for each source. Details and experimental comparison with the exact subset can be found in Appendix D.
>
> In line 161, we use the term “piecewise function” because there are two clearly different behaviors which depend on whether the dataset includes the entirety of Wikipedia. Compared to that, we believe the Math-500 figure has much more consistent trends.
>
> The variance reduction techniques are Maximal Marginal Relevance (MMR) and bagging. MMR induces more diverse documents, while bagging takes a different random subset of the top-10 for each trial.

---

### Official Review · Reviewer_fW4M · 2025-11-02

**Soundness:** 3
**Presentation:** 3
**Contribution:** 3
**Rating:** 6
**Confidence:** 3

**Summary:**

The paper asks how much useful signal in today’s open pre‑training corpora is left unexploited by pre‑training itself, and if it can be reclaimed at test time. Using a public 8B reader (Llama‑3.1 8B instruct) together with retrieval + reranking + self‑consistency yields +10.5 points on MMLU, +15.7 on Math‑500, +6.2 on GPQA, and 74% on SimpleQA. On MMLU, they fit a bounded sigmoid to the base models and report that adding retrieval behaves like a ~5× pre‑training compute multiplier on average (declining with scale).

**Strengths:**

- The design (pre‑train on a corpus, then retrieve from exactly that corpus at test time) probes how much of the corpus’ information is not captured parametrically. This bridges two active lines of work (RAG and inference‑time compute) in a controlled, data‑centric way.
- Paper does not hand‑wave contamination: it (a) quantifies overlap, (b) shows that decontaminated retrieval still helps (substantially for MMLU), and (c) surfaces the surprisingly large portion of benchmark content present in common open corpora
- “Inter‑document self‑consistency” diagnostic is clever and novel.
- Combination of retrieval, reranking, and self‑consistency is methodologically clean and yields strong additive gains

**Weaknesses:**

- MMLU has documented label and quality issues; using it as the sole basis to translate accuracy into compute multipliers risks reporting an artifact of the fit or of dataset flaws.
- Decontamination by token n‑gram overlap (16‑gram for MMLU, 26‑gram for Math‑500) is a good start but cannot remove paraphrastic or templatic leakage. Because the retrieval store is identical to pre‑training corpora, any residual overlap inflates the measured gap between “base” and “+retrieval.”
- The “compute multiplier” compares training FLOPs without charging anything to indexing, storage, retrieval/reranking inference, prompt expansion, or self‑consistency samples. In many deployments, inference cost dominates, and searching over such a large corpora is likely impractical at best and impossible at worst, due to the increased search compute cost. Without this analysis (of a break even point between inference and training flops), the multiplier is misleading operationally.
- The base models in §4 appear to be evaluated without retrieval but also without self‑consistency, whereas §5 compares an 8B reader baseline w/ CoT to retrieval + reranker + self‑consistency (+ MMR/bagging). While Table 5 includes a “self‑consistency only” baseline for the 8B reader, the earlier compute‑multiplier analysis relies on single‑pass answers for the base models. A more apples‑to‑apples comparison would allow multiple sampled chains and majority‑vote for the base models as well (possibly with no extra context) to isolate how much of the gain is from extra sampling vs. retrieval.
- The narrative proposes that retrieval may aid reasoning by “expanding context,” but the current evidence is correlational (for the stem benefits more than humanities claim). Are the largest per‑subject gains driven by questions where the retrieved passages contain the answer string (knowledge store) vs. passages that supply definitions/lemmas enabling reasoning (process aid)? A per‑question audit (string‑match / entailment / indirect support) would sharpen the mechanism story.

**Questions:**

- Add MMLU‑Pro/Redux, GSM8K/BBH (with grounding demands), and limited‑access tasks where retrieval should not help (negative control)
- How does the 5× multiplier change under (a) an unconstrained logistic fit; (b) a power‑law fit; and (c) fits trained only on the three middle scales (to reduce anchor effects)? Please report confidence intervals.
- Provide curves for k, reranker family, query rewriting, and inter‑document consistency at k>1 to situate the proposed VR/SC stack relative to standard RAG toolings

---

> ### Author Response · Authors · 2025-12-03
>
> Thank you for your time reviewing our work.
>
> While MMLU does have problems as you have pointed out, it is the most general pre-training evaluation that has been consistently used to measure pre-training model quality (especially in the open-source community). As observed in the SimpleQA numbers, different evaluations can lead to drastically different conclusions, but we stick with MMLU for the above reasons, and thus condition our numerical based compute multiplier claims on MMLU. Yet it is still fair to say that all of our evaluations see significant gains.
>
> Regarding decontamination, the datasets used for pre-training are never decontaminated, so the decontamination line in figure 1 only refers to the dataset once used for retrieval and thus shouldn’t inflate the gap.
>
> We agree with the reviewer that the current compute multiplier does not represent the full cost of deployment. To address these concerns, the indexing / embedding / reranking cost would be small relative to the pre-training budget because the embedding models are 0.6B parameters, orders of magnitude smaller than the reader model. On the other hand, consistency requires multiple inference passes with the reader model and could quickly dominate costs in a practical deployment. We note that whether the cost of deployment favors retrieval or not, our results still demonstrate that pretraining (dense) models is relatively inefficient at capturing the knowledge contained in the corpus, even at large flop budgets.
>
> The reviewer brings up a great point that Section 4 and 5 use different evaluation methods. This is due to the pre-trained models in Section 4 having slow inference, while Llama 3.1 8B is vLLM compatible while being relatively small in model size. Therefore the compute multiplier extrapolation from Section 4 to 5 is just an estimate, which we address in Section 5, though the baseline model performance in Section 5 that we use to anchor the compute multiplier calculation does use CoT.
>
> We agree that the evidence for “expanding context” is correlational, and our speculation here is to mention consistent patterns we observed. While we have not yet done a per-question audit, this is a great suggestion and something we will look to do for the next update. We will clarify this in the camera-ready, so as to avoid confusion.
>
> Unfortunately, we currently do not have the resources to run additional evaluations on MMLU-Pro, etc., but we will try to address the other questions:
>
> Here are the compute multipliers for MMLU (all) as requested, with the exception of unconstrained logistic which seemed unstable (highest budget compute ratio would be much higher than expected).
> | Type \ Budget | 5.64 x 10^21 | 1.90 x 10^22 | 7.04 x 10^22 | 1.74 x 10^23 | 7.34 x 10^23 | average | median |
> | ----- | ----- | ----- | ----- | ----- | ----- | ----- | ----- |
> | Baseline (constrained sigmoid) | 5.28 | 7.17 | 4.74 | 4.23 | 2.88 | 4.86 | 4.74 |
> | Power law | 5.91 | 8.94 | 5.27 | 3.80 | 1.67 | 5.12 | 5.27 |
> | Middle three points | 3.77 | 6.73 | 5.24 | 5.39 | 4.44 | 5.11 | 5.24 |
>
> To get confidence intervals (for the mean), we can resample from the test set and fit multiple (10k) curves and calculate compute multipliers for each. We get the following: constrained sigmoid - [4.38, 5.45], power law - [4.67, 5.63], middle three points - [4.26, 6.55]
>
> We can also include a table with the raw numbers in the Appendix for the final version, so that others may try different fits in the future.
>
> We choose not to use larger embedders / rerankers from the same family because the base models may be better than the reader model, and we do not experiment with query rewriting. Here are some additional k values for various strategies (including inter-document). We can include an expanded version with the other evaluations in the Appendix. Note that although inter-document performs better than self-consistency, it makes a lot more inference passes with the reader model.
> | Strategy | k=3 | k=6 | k=10 |
> | ----- | ----- | ----- | ----- |
> | MMLU w/ retrieval | 0.745 | 0.754 | 0.766 |
> | MMLU w/ reranker | 0.764 | 0.768 | 0.777 |
> | MMLU w/ reranker + self-cons. | x | x | 0.810 |
> | MMLU w/ reranker + inter-document | x | x | 0.813 |

---

### Official Review · Reviewer_8hXC · 2025-11-03

**Soundness:** 3
**Presentation:** 3
**Contribution:** 3
**Rating:** 6
**Confidence:** 4

**Summary:**

This paper studies how much existing LLM models utilize the pretraining data by letting models retrieve them during test time. The result shows that this can greatly improve the accuracy of various benchmarks, and thus imply the potential of the pretraining data might not be fully explored.

**Strengths:**

1. Their observation if able to be held to a larger scale might be interesting and could have some valuable guidance to the community.

**Weaknesses:**

1. The scale of the experiment is limited. It seems less surprising if smaller models could not fully utilize the pretraining data.
2. If the authors want to propose this as a method to be used rather than just an experiment for some observation, then they should compare it with other test time scaling methods.

**Questions:**

1. Could the authors show that the performance of existing off-the-shelf models can be improved by the proposed retrieval pipeline? This does not require pretrain so the author should be able to experiment with larger models. This could also rule out the possibility that the models used in the experiment might not be pretrained properly.

---

> ### Author Response · Authors · 2025-12-03
>
> Thank you for your time reviewing our work.
>
> Regarding experiment scale, we believe that our largest pre-training experiments at ~7e23 FLOPs are within 1-2 orders of magnitude of many frontier training runs (e.g. Deepseek v3 at 3e24 FLOPs), albeit dense. We could overtrain the models, however Chinchilla and related scaling laws would suggest that this would be less compute efficient and favor retrieval even more.
>
> The reviewer makes a good point that off-the-shelf models may have different behavior. It would be difficult to replicate this experiment because of the lack of knowledge regarding pre-training data used to train them. While some benchmarks like SimpleQA will benefit from retrieval as it is highly likely they also train on Wikipedia, frontier models already do quite well on MMLU and this is likely due to a superior pre-training mix, possibly in combination with MoE-style sparsity and other techniques to improve data-efficiency.

---

### Meta-Review · Area_Chair_TUq9 · 2026-01-07

**Summary:**

This paper investigates whether pre-training data can be more effectively utilized through retrieval augmented generation (RAG) at test time. This is a very interesting idea, and the authors pre-train models at various scales and demonstrate that retrieval from the same pre-training data provides significant performance gains, acting as a ~5x compute multiplier for MMLU. They argue this suggests pre-training methods underutilize available data, hinting at potential ideas for improved architecture or training algorithms. The work received mostly positive reviews, with consensus that the research question is interesting but some minor concerns remain. Overall, I think the paper explored a novel idea with sufficiently extensive experiments to support the findings.

**Reviewer Concerns:**

The rebuttal successfully addressed several presentation and clarity issues. Authors provided clear explanations of their RAG methodology, clarified which models were used in each experiment, and provided bootstrap confidence intervals for compute multipliers. They gave reasonable justifications for focusing on same-dataset retrieval and explained experimental design differences between sections due to inference speed constraints. Authors also provided partial FLOP calculations showing embedding/reranking costs are <1% of pre-training, and clarified that decontamination only applies to retrieval data, not pre-training data.

Some methodological concerns, though less critical, remain unresolved such as the compute multiplier calculation issue, the validity of the "underutilization" claim, missing comparisons with other test-time scaling methods, etc.

**Reviewer Scores:**

Most of the reviewers are positive, and one reviewer likely moving from weak accept to accept due to improved clarity.

---

### Decision · Program_Chairs · 2026-01-26

Accept (Poster)